# Evidence of Rift Valley Fever Virus Circulation in Livestock and Herders in Southern Ghana

**DOI:** 10.3390/v15061346

**Published:** 2023-06-10

**Authors:** Sherry Ama Mawuko Johnson, Richard Asmah, Joseph Adongo Awuni, William Tasiame, Gloria Ivy Mensah, Janusz T. Paweska, Jacqueline Weyer, Orienka Hellferscee, Peter N. Thompson

**Affiliations:** 1Department of Production Animal Studies, Faculty of Veterinary Science, University of Pretoria, Onderstepoort 0110, South Africa; 2School of Veterinary Medicine, University of Ghana, Legon, Accra 00233, Ghana; 3School of Biomedical & Allied Health Sciences, University of Ghana, Accra 00233, Ghana; rhasmah@ug.edu.gh; 4Accra Veterinary Laboratory, Ministry of Food and Agriculture, Accra P.O. Box M161, Ghana; josephawuni@hotmail.com; 5School of Veterinary Medicine, Kwame Nkrumah University of Science and Technology, Kumasi 00233, Ghana; drwilly2002@gmail.com; 6Noguchi Memorial Institute for Medical Research, University of Ghana, Accra 00233, Ghana; gmensah@noguchi.ug.edu.gh; 7Centre for Emerging Zoonotic and Parasitic Diseases, National Institute for Communicable Diseases of the National Health Laboratory Service, Sandringham, Johannesburg 2192, South Africa; januszp@nicd.ac.za (J.T.P.); jacquelinew@nicd.ac.za (J.W.); orienkah@nicd.ac.za (O.H.); 8Department of Medical Virology, Faculty of Health Sciences, University of Pretoria, Pretoria 0002, South Africa; 9Department of Medical Virology, Faculty of Health Sciences, University of Witwatersrand, Johannesburg 2050, South Africa

**Keywords:** zoonosis, Rift Valley fever, Ghana, One Health, vector-borne disease

## Abstract

Rift Valley fever (RVF) is a re-emerging zoonotic disease of domestic ruminants and humans. While neighbouring countries have reported outbreaks of RVF, Ghana has not yet identified any cases. The aim of this study was to determine whether RVF virus (RVFV) was circulating in livestock and herders in the southern part of Ghana, to estimate its seroprevalence, and to identify associated risk factors. The study surveyed 165 livestock farms randomly selected from two districts in southern Ghana. Serum samples of 253 goats, 246 sheep, 220 cattle, and 157 herdsmen were tested to detect IgG and IgM antibodies against RVFV. The overall seroprevalence of anti-RVF antibodies in livestock was 13.1% and 30.9% of farms had RVFV seropositive animals. The species-specific prevalence was 24.1% in cattle, 8.5% in sheep, and 7.9% in goats. A RVFV IgG seroprevalence of 17.8% was found among the ruminant herders, with 8.3% of all herders being IgM positive. RVFV was shown, for the first time, to have been circulating in southern Ghana, with evidence of a recent outbreak in Kwahu East; however, it was clinically undetected despite significant recent human exposure. A One Health approach is recommended to better understand RVF epidemiology and socio-economic impact in Ghana.

## 1. Introduction

Rift Valley fever (RVF) is a mosquito-borne viral zoonotic disease caused by the Rift Valley fever virus (RVFV), a phlebovirus of the family *Phenuiviridae* [1]. It is considered a re-emerging disease of significant risk to public health, socio-economics, and trade [2]. It causes non-specific and mild febrile illness in humans and can result in complications such as retinitis, hepatitis, and haemorrhagic and neurological syndromes [3,4,5,6]. Ruminant herders, abattoir workers, and veterinarians are among those at the greatest risk of exposure and infection [7]. In countries where the disease occurs, RVFV infections are considered underdiagnosed and widespread among people in close occupational contact with livestock [8].

Rift Valley fever causes high mortality among young animals and/or abortion in adult livestock, irrespective of the stage of gestation [9]. In pregnant sheep, mortality could be up to 100% and 60% in adult ruminants [10].

Rift Valley fever has been reported in many West African countries, including Nigeria [11], Niger [12], Cameroon [13], and two countries bordering Ghana, namely, Cote d’Ivoire [14] and Burkina Faso [15,16]. Ghana has a warm and humid climate and experiences perennial flooding in the south, particularly in Accra and its environs, due to rapid urbanization, climate variability, and flaws in physical planning [17,18,19]. These conditions favour vector activity, a key factor for transmission and dissemination of RVFV.

At present, RVF is not one of the priority scheduled diseases for active surveillance in Ghana by the Veterinary Services of the Ministry of Food and Agriculture [20] or the Ghana Health service [21]. Consequently, RVF will not be considered among the list of diseases causing abortions in livestock and febrile and haemorrhagic cases in humans since the disease was not known to be found in Ghana. In 2016, the European Centre for Disease Prevention and Control (ECDC) reported a case of RVF in an adult male in France that “originated probably in Ghana” [22]. Given the zoonotic nature of the disease and the clinical presentation in humans, RVF could be mistaken for malaria or any other febrile illness. Therefore, a survey to determine its presence and level of occurrence in livestock and humans in Ghana is key to protecting human and animal health.

A serological result of a small number of stored sheep and goat serum samples at the Accra Veterinary Laboratory in 2010 [23] suggested that RVFV may have been circulating among livestock in Ghana. We sought to confirm the occurrence and determine the seroprevalence of RVFV in domestic livestock and livestock farmers in the southern zone of Ghana and to assess associated risk factors for its occurrence to facilitate the detection, control, and prevention of both livestock and human infection.

## 2. Materials and Methods

### 2.1. Study Area

The study was conducted between July 2019 and February 2020 in two southern regions of Ghana, which is bordered on the west by Cote d’Ivoire, the north by Burkina Faso, the east by Togo, and the south by the Atlantic Ocean and Gulf of Guinea. Ghana has 16 administrative regions and an estimated population of 30.8 million as of 2021 [24]. The country has different ecological zones with tropical forests and coastal savannahs in the south, humid savannahs in the centre, and dry savannahs in the northern belts [18]. The climate is warm and humid with a bi-modal rainfall cycle in the south and central belts and is uni-modal in the northern belt. The southern and central belts experience major rainfall in March to July and a minor rainy season in September to November [18].

The study locations were Ga South and Kwahu East in the Greater Accra and Eastern regions, respectively. Ga South municipality is one of the 26 municipal, metropolitan, district assemblies (MMDAs) in the Greater Accra region, a coastal savannah agro-ecological zone, and one of the flood-prone areas during heavy rainfall [17]. It includes three vegetation covers: moist semi-deciduous forest, mangrove swamp, and coastal scrub and grassland [25]. The municipality is in the southwestern part of Accra, occupies a land area of 517.2 square kilometres [26], and has a population of 350,121 [24] with 362 communities, mainly engaged in agriculture and fishing.

The Kwahu East district is situated in the northern part of the Eastern region, and it is known for its undulating landscape with steep slopes, several rock outcrops, and scarps. The height of mountain peaks ranges between 220 and 640 m above sea level [27]. Kwahu East is considered to have a wet semi-equatorial climatic zone and a population of 79,726 inhabitants [26]. The district was selected for being in a semi-equatorial zone with forest cover.

### 2.2. Target Population and Sample Size

The target livestock were domestic sheep, goats, and cattle. In the absence of a published data on the prevalence of RVF in Ghana, a prevalence of 20% reported in Burkina Faso [17] was used as an estimate for sample size calculation. Burkina Faso neighbours Ghana on the north and has similar ecological zones. With an estimated prevalence of 20%, a 95% confidence interval, and a maximum allowable error of 5%, the sample size was computed to be 246 ruminants. The obtained sample size was multiplied by a design effect (D) of 3 using an intracluster correlation coefficient (rho) of 0.25 and an average cluster size of 9 based on the formula D = 1 + p (m − 1) [28]. This gave a total sample size of 738 ruminants for the study (246 of each species).

### 2.3. Sampling-Farms and Livestock

A two-stage sampling method was employed. In stage one, the list of livestock farms in the selected districts was obtained from the local district veterinary offices. The farm lists constituted the sampling frame from which farms were randomly selected. On multi-species farms, up to 9 sheep, goats, and cattle were sampled using systematic random sampling. On single-species farms, the available ruminant type was sampled, and another farm within the sampling frame was selected for the other two species to make up for the sample size of the other species.

### 2.4. Variables Measured

A structured questionnaire was administered to the livestock farmers to assess demographic data of livestock (age, sex, breed), farm characteristics (farm size, type of housing, proximity to water bodies), and flock health history (history of abortion, disposal of birthing and aborted materials, clinical signs at the time of sampling).

### 2.5. Blood Sampling and Laboratory Analyses

Five mL of blood was drawn from the jugular vein directly into an 8.5 mL Vacutainer^®^ tube with clot-activator and gel for serum separation. Sera were transported on ice to a biosafety level 2 laboratory at the Noguchi Memorial Institute for Medical Research of the University of Ghana, where they were centrifuged at 2500 rpm for 2–3 min and aliquoted. The aliquot in cryogenic vials were stored at −20 °C until ready for testing. The sera were tested for anti-RVFV antibodies using a commercial competitive enzyme-linked immunosorbent assay (ELISA, ID Screen^®^ Rift Competition Multi-Species ELISA, IDVet, rue Louis Pasteur, France). The sensitivity and specificity of the test were documented to be 91–100% and 100%, respectively [29]. All samples that tested positive were then also tested using the RVF IgM Capture ELISA (IDVet, France) in order to determine recent viral infection. The ELISA tests were performed according to the manufacturer’s instruction.

### 2.6. Human Serosurvey

The human serosurvey targeted livestock farmers or herders who worked directly with ruminants in the study areas. On each livestock farm included in the study, the herder or any close relatives who tendered livestock and consented were purposively selected and, after giving informed consent, were sampled. A maximum of 5 mL of blood was collected from each consenting farmer or herder. A structured questionnaire was administered to participants to obtain socio-demographic information (age, gender, level of education, occupation), as well as potential risk factors such as exposure to mosquitoes, contact with livestock, use of protective clothing when handling sick animals, type of housing, and consumption of raw milk.

Field assistants were drawn from the study region due to their familiarity with the research area. They were trained on the correct administration of the questionnaire to ensure consistency of responses and to reduce interview-related errors. The questionnaire was pre-tested in a district other than the selected one for the study and administered by trained field assistants.

Human sera were shipped to the Centre for Emerging Zoonotic and Parasitic Diseases, National Institute for Communicable Diseases, Johannesburg, South Africa. The samples were triple packed and shipped in accordance with CDC’s prescribed guidelines for packaging and transporting infectious substances [30]. The samples were tested using inhibition ELISA for the detection of RVFV antibodies; this assay did not distinguish between IgG and IgM [31] ]. All positive samples by the inhibition ELISA were re-tested using IgG-sandwich ELISA to detect anti-RVFV IgG and IgM-capture ELISA to detect anti-RVFV IgM, as described previously [32,33].

### 2.7. Data Analysis

Seroprevalence of RVFV was estimated overall and for each species, with corresponding 95% confidence interval (CI) using the logit transformation with farm-specific cluster robust standard error. We assessed the bivariate relationship between each categorical exposure variable and outcome (RVFV seropositivity) using the Chi-square test of independence (and Fisher’s exact test where cell counts were low), followed by a multivariable modified Poisson regression model with robust standard error that reported prevalence ratios and corresponding CI. Separate models were conducted for livestock (with species as an explanatory variable) and for humans. In the latter, in order to assess the association between livestock and human seropositivity, the presence of at least one seropositive animal on the farm was included as an explanatory variable. For initial variable selection, we used a significance level of 20% followed by stepwise backward selection until all variables were significant at *p* ≤ 0.05. All statistical analyses were conducted in Stata SE version 16 (StataCorp, College Station, TX, USA).

### 2.8. Ethics Clearance

Ethics clearance was obtained from the Institutional Committee of Animal Use and Care, University of Ghana (UG-IACUC 010/18-19), Faculty of Veterinary Science, University of Pretoria Research Ethics Committee (REC065-19) and University of Pretoria Animal Ethics Committee (REC065-19). Livestock Farmers’ Associations in the selected regions and districts were consulted. Permission was granted by the field veterinarians in the study districts. Individual written informed consent was sought and obtained from the selected livestock farmers and herders after verbal explanation of the aims of the study, and participation was voluntary.

For human sampling, ethics clearance was obtained from Research Ethics Committee, Faculty of Health Sciences, University of Pretoria (554/019). Additionally, clearance was obtained from the Ethics and Protocol Review Committee of the School of Biomedical and Allied Health Sciences, University of Ghana (SBAS-MLS. /SA/2018-2019).

## 3. Results

### 3.1. Farm Characteristics

A total of 251 sheep, 252 goats, and 216 cattle was sampled from 165 farms, 32 in the Ga South and 133 in the Kwahu East districts. Most of the farms (131; 79.3%) practiced semi-intensive farming system, where animals were grazed during the day and penned during the night. The rest of the farms practiced extensive (29; 17.6%) and intensive (5; 3.0%) systems of farming. The farms were mixed species (67; 40.6%) of sheep, goats, and cattle. Others were single-species, 24 (14.6%) with cattle, 38 (23.0%) with goats, and 36 (21.8%) with sheep. Most farms (139; 84.2%) were situated near water bodies and in flood-prone areas in both districts, as shown in Figure 1 and Figure 2.

### 3.2. Breeds and Flock Sizes

Livestock kept at the time of sampling were mostly females (73.8%). Their ages ranged from 5 to 144 months with a median of 30. About half of the farms (51.7%) had reported abortions in recent months (median period = 6 months; range 1–36 months) prior to sampling. However, abortion rates could not be calculated, as farm records of health were poorly kept.

Cattle breeds sampled were N’damas (36.1%), Gudali (23.6%), West Africa short horn (20.4%), Zebu cross (18.1%), and Sangas (1.8%). The sheep were mainly Djallonkes (61.8%) and Sahelian (38.2%). Most of the goats were West Africa dwarfs (75.8%) and Sahelian (24.2%).

### 3.3. Seroprevalence in Ruminants

The distribution of RVFV seropositivity is presented in Table 1. The overall seroprevalence of RVF in the ruminants was 13.1% [95% CI: 10.5–16.2]. The species-specific prevalence was 24.5% [95% CI:18.9–30.8] in cattle, 8.4% [95% CI: 5.3–12.5] in sheep, and 7.9% [95% CI: 4.9 11.9] in goats. Evidence of recent RVFV infection (positive for RVFV IgM) was found in four (0.6%) of the livestock, three from Ga South and one from Kwahu East.

### 3.4. District and Farm-Specific Prevalence of RVFV

A total of 47 farms (28.5% [95% CI 22.3–36.6]) had RVFV seropositive animals. At the district level, the proportion of farms with positive animals was 28.1% for Ga South and 31.7% for Kwahu East districts. Farm level prevalence ranged from 0 to 85.5% with a median of 10.0%. The district specific prevalence was 6.4% and 15.4% for Ga South and Kwahu East, respectively (Table 1).

Table 1 also shows the bivariate analysis of factors associated with RVF infection in the animals. District, species, age, sex, and feeding practice were selected for the multivariate model. Multivariate analysis of factors associated with RVFV seropositivity in livestock is presented in Table 2. The results from the modified Poisson regression model showed that the prevalence of RVF infection in livestock was higher in Kwahu East than Ga South after adjusting for confounding, lower in sheep and goats than in cattle, and lower in farms where grazing and mixed feeding were practiced compared to cut and carry only.

### 3.5. Seroprevalence in Ruminant Herders

A total of 157 herders were tested from 117 farms (Table 3). The median age of the respondents was 44 years (range 18–66 years). A total of 25 farms (21.3%) had RVFV seropositive herders. None of the herders tested exhibited signs suggestive of ill health at the time of sampling. The overall seroprevalence of anti RVFV antibodies in the herders was 17.8% [95% CI 12.2–24.7] using both inhibition ELISA and IgG sandwich ELISA, and the two results agreed completely. The overall IgM seroprevalence was 8.3% (13/157) and, of the 28 seropositive herders, 46% (13/28) were IgM positive, evidence of recent infection. All 13 of the IgM seropositive herders were from the Kwahu East district.

The only factors significant in the multivariable model (Table 4) were district (prevalence was seven times higher in Kwahu East than Ga South) and gender (higher in males than females).

Figure 3 shows the distribution of farms with RVFV-positive livestock and herders in the study sites. The seropositive farms appear clustered around the Lake Volta and Weija dam in the Kwahu East and Ga South districts, respectively. Several areas (red and black balls) are shown to farms where both herders and livestock were exposed to RVFV. Sempoa, a rural and less-privileged community in the Kwahu East had the highest proportion of farms (24.2%) with infected ruminants and herders. Livestock health history in Sempoa showed that many cattle had died suddenly in 2018; however, these deaths were not investigated. The community was without an animal health post and relied on animal health staff from other distant communities.

## 4. Discussion

This study provides the first conclusive evidence of RVFV circulation in Ghana, both in domestic ruminant species and in humans. Serological evidence of infection was found in both Ga South in the Greater Accra region and Kwahu East in the Eastern region of Ghana. The detection of IgM in both livestock and herders, notably in Kwahu East, indicates that a recent, undetected RVF outbreak has occurred, likely causing undiagnosed human and livestock illness. As IgM is a known marker of recent infection and generally declines 2–3 months post-infection in viral infections [34], exposure of herders in Kwahu East to RVFV within the previous few months was demonstrated.

An overall RVFV prevalence of 18% in livestock and 13% in humans likely has significant public health implications. Rift Valley fever presents generally as a febrile and influenza-like illness and could easily be mistaken for malaria and treated as such. Among the herders tested, most presented apparently healthy, consistent with the fact that RVFV could be missed or mistaken for another illness [35]. This was consistent with a study in Mozambique where RVFV-seroconverting patients were misdiagnosed and assumed to be malaria cases [36]. This situation may also be the case in Ghana, as malaria is endemic and remains amongst the top five differential diagnoses for febrile illness in humans.

Our findings in both ruminants and herders were consistent with those reported from areas in which RVFV was known to circulate endemically with or without periodic outbreaks in East Africa [37], West Africa [38], and South Africa [35]. Several studies have indicated circulation of RVFV in West Africa [14,38], some with subclinical circulation and others with outbreaks in livestock.

The fact that almost half of the herders in Kwahu East who tested positive for anti RVFV IgG were also positive for anti RVF IgM indicates a recent undetected outbreak in Kwahu East. This is supported by the fact that RVFV seroprevalence is high in all age groups of livestock in Kwahu East. In contrast, seroprevalence in livestock in Ga South showed seroprevalence gradually increasing with age, being highest in the oldest animals. This suggests low-level endemic circulation in that area, in which seasonally flooded wetlands provide suitable conditions for maintenance of RVFV circulation in mosquito vectors.

The highest RVFV seroprevalence in livestock (15%) was recorded in Kwahu East, more than twice the prevalence recorded in Ga South (6%). This was to be expected following an outbreak, with high seroprevalence in all age groups. Livestock farmers lived in close proximity with livestock in Kwahu East and were likely to be exposed to infected animals (including birthing materials) in addition to the vector activity. However, no association was found between human RVFV seropositivity and livestock seropositivity on the same farm. Although human exposure was thought to occur mainly via exposure to infected animals, our results could indicate significant human exposure due to mosquito bites. This requires further investigation.

As many as 29% of the farms sampled had seropositive livestock, showing widespread occurrence of RVFV in the study area. Less than 1% of the seropositive livestock showed evidence of recent infection, and this agrees with similar findings where very recent animal outbreaks have not occurred [39]. The farm-level prevalence could not be ascribed to vaccination, as Ghana has never vaccinated livestock against RVFV, particularly because it was not thought to be present in the country.

In Kwahu East, RVFV seropositivity was clustered around Sempoa and Hywohoden (Figure 3). These two communities were noted for Fulani-herder and nomadic activities resulting in conflicts, as reported in recent occurrence in southern Ghana [40]. The nomadic Fulani herdsmen were reported to be from Mali, Niger, Nigeria, and other undetermined places who moved in and out of these communities with their livestock for greener pastures. According to FAO [41], animal movements, trade, and weather conditions were the main risk factors for the occurrence and reoccurrence of RVF in West Africa and spread to unaffected areas.

It was not possible to ascertain abortion rates in this study due to poor record keeping of health history on the farms. The source of replacement stock was found to be from within Ghana. However, the original source of the livestock sold in the open market was difficult to ascertain, as livestock mobility through transhumant activities was difficult to record. Livestock movements are considered an important driver of infectious disease transmission and spread, and the many unapproved routes of transporting livestock to and from Ghana could be a risk factor for the spread of diseases. Further study is required to establish the role of transhumance in the spread of transboundary animal diseases such as RVF in Ghana.

In a study to map seasonally varying environmental suitability of RVF in Africa, Ghana was found to be suitable for RVFV outbreaks [42]. Given that livestock farming is the second income-generating activity in the Kwahu East district [43], a large outbreak of RVF could be devastating to the livelihoods of the population, and preventative measures are required. Our finding, therefore, calls for RVFV to be kept on the radar of the animal and human health sectors in Ghana.

## 5. Conclusions

This study indicated that RVFV has been circulating in livestock and humans in the southern part of Ghana. Kwahu East showed a higher level of RVFV exposure in livestock and humans compared to Ga South, with evidence of recent infection, indicating an undetected outbreak in livestock and humans. Further studies are recommended to understand the burden of RVF in Ghana. A One Health approach is required to further investigate the epidemiology and socioeconomic impact of RVF and for risk mapping to predict and prevent outbreaks.

## Figures and Tables

**Figure 1 viruses-15-01346-f001:**
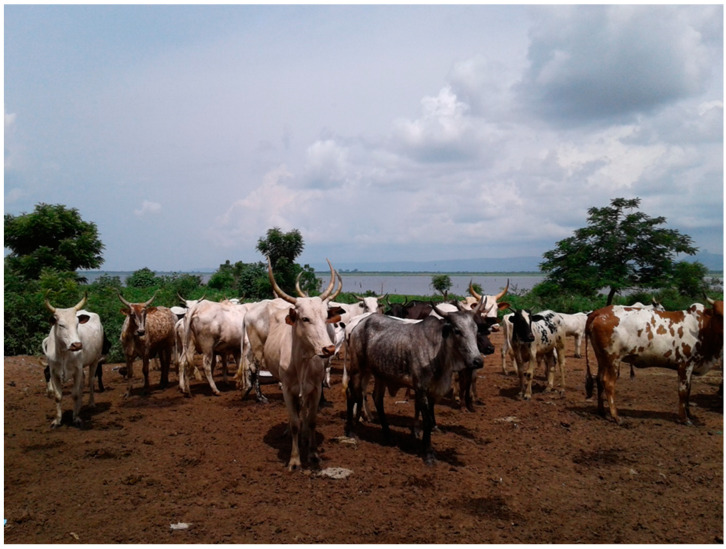
A cattle kraal situated less than 200 m from the Afram River in Kwahu East of the Eastern region of Ghana. Picture by S. Johnson.

**Figure 2 viruses-15-01346-f002:**
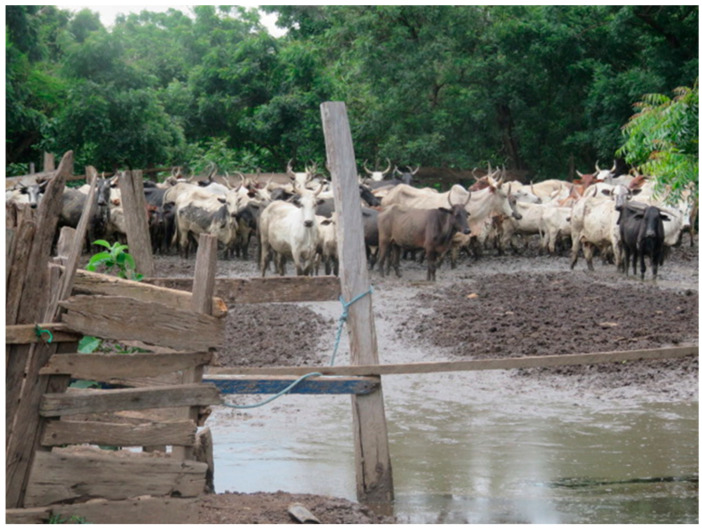
A flooded cattle kraal at Hyewohoden in the Kwahu East District, Eastern Region of Ghana. Picture by S. Johnson.

**Figure 3 viruses-15-01346-f003:**
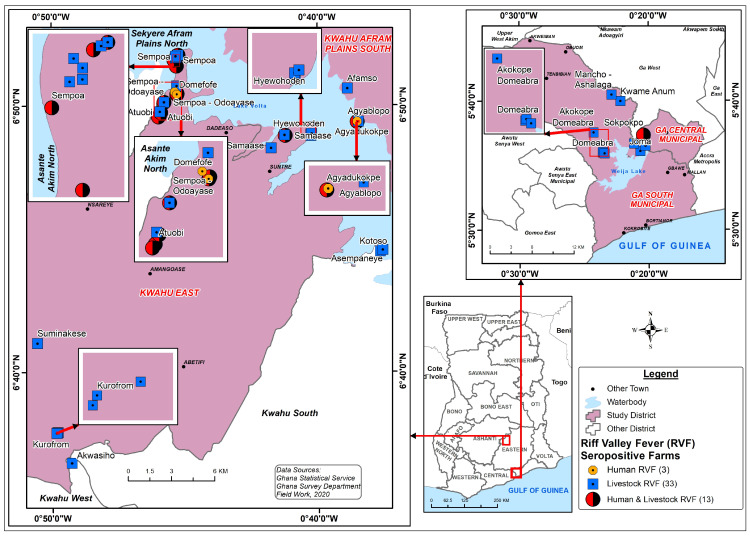
Distribution of RVFV seropositive farms and herders Ga South and Kwahu East, Ghana.

**Table 1 viruses-15-01346-t001:** Prevalence and bivariate analysis of factors associated with Rift Valley fever seropositivity in domestic ruminants in Kwahu East and Ga South districts, Ghana.

Factor		N	Overall Prevalence (%)	*p*-Value	District-Specific Prevalence					
					Ga South			Kwahu East		
					*n*	Prevalence (%)	*p*-Value	*n*	Prevalence (%)	*p*-Value
All species		719	13.1		187	6.4		532	15.4	
Species				0.001			0.003			<0.001
	Cattle	216	24.5		87	12.6		129	32.6	
	Goat	252	7.9		45	2.2		207	9.2	
	Sheep	251	8.4		55			196	10.7	
Farm type				0.314			0.054			0.323
	Mixed	304	11.5		100	8.0		204	13.2	
	Single-species	415	14.2		87	4.6		328	16.8	
Feeding practice				<0.001			0.054			<0.001
	Cut and carry	52	17.3		15	6.7		37	21.6	
	Grazing	333	18.0		100	10.0		233	21.5	
	Mixed	334	7.5		72	1.4		262	9.1	
Sleeping place at night				0.453			0.005			0.489
	Enclosed	528	12.5		130	3.1		398	15.6	
	Open field	191	14.7		57	14.0		134	14.3	
Sex				0.032			0.541			0.150
	Female	531	14.7		119	7.6		412	16.8	
	Male	188	8.5		68	4.4		120	10.8	
Age (months)				0.022			0.019			0.378
	≤18	213	9.4		88	3.4		125	13.6	
	19–29	107	8.4		21	0.0		86	10.5	
	30–39	177	14.1		44	6.8		133	16.5	
	≥40	222	18.0		34	17.7		188	18.1	
Husbandry practice				0.463			1.000			0.954
	Extensive	106	16.0		0	-		106	16.0	
	Semi intensive	597	12.7		179	6.7		418	15.3	
	Intensive	16	6.3		8	0.0		8	12.5	

**Table 2 viruses-15-01346-t002:** Multivariable analysis of factors associated with Rift Valley fever seropositivity in ruminants in Kwahu East and Ga South districts, Ghana.

Factor		Prevalence Ratio [95% CI]	*p*-Value
District			
	Ga South	1 *	
	Kwahu East	3.2 [1.8–5.5]	<0.001
Species			
	Cattle	1 *	
	Goat	0.3 [0.1–0.4]	<0.001
	Sheep	0.3 [0.2–0.5]	<0.001
Feeding			
	Cut and carry	1 *	
	Grazing	0.5 [0.2–0.9]	0.024
	Mixed	0.3 [0.1–0.6]	0.002

* Reference level.

**Table 3 viruses-15-01346-t003:** Bivariate analysis of factors of Rift Valley fever virus seropositivity among herders in the Kwahu and Ga South districts, Ghana.

Factor		*n*	RVFV Seroprevalence (%)	*p*-Value
District				0.015
	Ga South	32	3.1	
	Kwahu East	125	21.6	
Gender				0.050
	Female	53	9.4	
	Male	104	22.1	
Age				0.860
	≤30	18	22.2	
	31–40	45	17.8	
	41–50	53	15.1	
	≥50	41	19.5	
Assist in delivery				0.869
	No	80	16.3	
	Yes	77	19.5	
Abortion on farm				0.828
	No	56	16.1	
	Yes	101	18.8	
Live near livestock				0.839
	No	70	17.1	
	Yes	87	18.4	
Presence of flies on farm				0.363
	No	7	28.6	
	Yes	150	17.3	
Farm RVFV status				0.837
	Negative	82	17.1	
	Positive	75	18.7	

**Table 4 viruses-15-01346-t004:** Multivariable analysis of factors associated with Rift Valley fever virus seropositivity among herders in Kwahu and Ga districts, Ghana.

Factor	Prevalence Ratio [95% CI]	*p*-Value
District		
Ga South	1 *	
Kwahu East	7.5 [1.1–52.8]	0.043
Gender		
Female	1 *	
Male	2.5 [1.0–6.3]	0.041

* Reference level.

## Data Availability

The data presented in this study are available on request from the corresponding authors. The data are not publicly available due to ethical reasons.

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
