# Peer review of "Evidence of Rift Valley Fever Virus Circulation in Livestock and Herders in Southern Ghana"

_viruses, 2023, doi:10.3390/v15061346_

Round 1

Reviewer 1 Report

The manuscript entitled “Evidence of Rift Valley fever virus circulation in livestock herders in Southern Ghana” by Johnson S.A.M., et al. described that IgG and IgM specific to RVFV N protein were detected in livestock animals and humans in two districts in Ghana. Serological study indicated the circulation of RVFV among livestock animals (sheep, cattle, goats) and herders in Sothern Ghana. Overall, this manuscript is well written, while providing valuable information regarding the seroplevanence of anti-RVFV antibodies in animals and humans in Ghana. Some minor points are raised to strengthen the content.

Minor points:

·         Positive results of IgM or IgG ELISA do not exclude cross-reactivity to other phleboviruses (e.g., Arumowot, Gabek Forest and Gordil viruses in west Africa). It is recommended to include further discussion about the possible limitation of ELISA. If it is possible, additional testing of selected positive and negative sera by the plaque reduction neutralizing test will strengthen the evidence of overall survey.

·         Line 117: “single-species”

·         Line 156: “by Paweska” may be changed to “as described previously”.

·         Line 189: “17.6%”?

·         Line 222: “8.4%”? (please be consistent with Table 1)

·         Line 226: “A total of 47 farms (30.9%; [95% CI 23.6 -38.9]) had RVFV seropositive animals.” In this sentence, how could 30.9% be calculated? Total farm number was 165.

Author Response

Please find attached our response to the comments. Thank you

Reviewer 2 Report

Rift Valley fever (RVF) is a re-emerging mosquito-borne viral zoonotic disease of domestic ruminants and humans. Since possible circulation of RVFV among livestock in Ghana was reported in 2010, authors aimed to confirm the RVFV circulation in domestic livestock and livestock farmers in the southern part of Ghana and to estimate its seroprevalence and to identify associated risk factors. This is the honest paper worth publishing. However, I suggest certain things, which need attention, improvement and clarification to support and strengthen the overall impact of the article.

Major points for attention:

It is necessary to provide data about storing conditions of collected livestock samples, if they weren’t processed immediately after preparation, also authors indicated that sera samples were centrifuged but they didn’t indicate at what conditions (centrifugation time and force). It I also necessary to indicate shipping conditions of collected human sera samples, their preparation for ELISA testing and storing conditions of prepared samples. In line 153 authors indicated that they used inhibition ELISA for RVFV antibodies detection but they didn’t specify which ELISA test was used. Authors should indicate whether the ELISA tests were performed according manufacturer’s instructions ore some changes were made. It should be indicated how the microplates were read and what optical density is recorded and what kind of ELISA processor was used. Advantage of used ELISA tests should be indicated also.

Minor points:

I suggest to move the subchapters 3.1. and 3.2 into Materials and Method section.

References should be described according to the Viruses citing style.

Journal Articles:

1. Author 1, A.B.; Author 2, C.D. Title of the article. Abbreviated Journal Name Year, Volume, page range (without p. or pp.).

Websites:

Title of Site. Available online: URL (accessed on Day Month Year).

Title of Site. URL (archived on Day Month Year).         

Author Response

Please see that attachment
